# A phenotypic and genomics approach in a multi-ethnic cohort to subtype systemic lupus erythematosus

Cristina M. Lanata [1,6], Ishan Paranjpe[2,3,6], Joanne Nititham[1], Kimberly E. Taylor[1], Milena Gianfrancesco[1], Manish Paranjpe[2], Shan Andrews[2], Sharon A. Chung[1], Brooke Rhead[4], Lisa F. Barcellos[4], Laura Trupin[1], Patricia Katz[1], Maria Dall'Era[1], Jinoos Yazdany[1], Marina Sirota[2,6] & Lindsey A. Criswell[1,5,6]

Systemic lupus erythematous (SLE) is a heterogeneous autoimmune disease in which outcomes vary among different racial groups. Here, we aim to identify SLE subgroups within a multiethnic cohort using an unsupervised clustering approach based on the American College of Rheumatology (ACR) classification criteria. We identify three patient clusters that vary according to disease severity. Methylation association analysis identifies a set of 256 differentially methylated CpGs across clusters, including 101 CpGs in genes in the Type I Interferon pathway, and we validate these associations in an external cohort. A cis-methylation quantitative trait loci analysis identifies 744 significant CpG-SNP pairs. The methylation signature is enriched for ethnic-associated CpGs suggesting that genetic and non-genetic factors may drive outcomes and ethnic-associated methylation differences. Our computational approach highlights molecular differences associated with clusters rather than single outcome measures. This work demonstrates the utility of applying integrative methods to address clinical heterogeneity in multifactorial multi-ethnic disease settings.

[1] Russell/Engleman Rheumatology Research Center, Department of Medicine, University of California San Francisco, San Francisco, CA, USA. [2] Bakar Computational Health Sciences Institute, University of California, San Francisco, CA, USA. [3] Icahn School of Medicine at Mount Sinai, New York, NY, USA. [4] University of California, Berkeley, CA, USA. [5] Institute for Human Genetics, University of California, San Francisco, San Francisco, CA, USA. [6] These authors contributed equally: Cristina M. Lanata, Ishan Paranjpe, Marina Sirota, Lindsey A. Criswell. Correspondence and requests for materials should be addressed to L.A.C. (email: Lindsey.Criswell@ucsf.edu)

Systemic lupus erythematosus (SLE) is a multifactorial autoimmune disease with heterogeneous manifestations that encompasses a wide range of disease severity. Complex diseases such as SLE involve a dynamic interplay between molecular processes, many of which are unknown. Long-term outcomes for individual patients are therefore difficult to predict, as is the scope of organ system involvement. While some patients have aggressive disease progression, others do not accrue significant damage within 5 years of SLE diagnosis[1–4]. We know little about why an affected individual might develop a particular SLE phenotype. Furthermore, a patient can be classified as having SLE if she or he fulfills any four of the 11 American College of Rheumatology (ACR) classification criteria[5], with resultant extensive disease heterogeneity. In recent years, significant effort has been applied to better sub-classify SLE, not only to predict disease outcomes but also inform specific mechanistic pathways that could be strategically targeted according to subtype[6–8].

SLE disease progression and outcomes vary significantly among different racial/ethnic groups[9–11]. Patients from non-European populations, such as Hispanics, African Americans, and Asians, develop SLE at a younger age and experience worse disease manifestations than patients of European descent. Even after decades of basic research and public health initiatives these health disparities remain relatively unchanged. Factors that underlie these disparities are elusive and likely derive in part from complex interactions between genetic and environmental factors, which may in part originate from social inequities. However, the majority of molecular studies to date have been carried out in European populations.

These is evidence that both genetics and DNA methylation play a role in SLE outcomes. Lupus nephritis, a severe outcome of lupus that drives disease mortality, was found to be significantly correlated with genetic variants in *ITGAM*, *TNFSF4*, *APOL1*, *PDGFRA*, and *SLC5A11*, among others. The *HLA-DR2* and *HLA-DR3* alleles have also been associated with susceptibility and autoantibody production in lupus[12–15]. Overall, hypomethylation of interferon-responsive genes has been associated with higher disease activity and renal disease, as well as production of autoantibodies[16–18]. For example, differentially methylated CpGs in *TNK2*, *DUSP5*, *MAN1C1*, *PLEKHA1*, *IRF7*, *HIF3A*, *IFI44*, and *PRR4* have been associated with lupus nephritis[18–20]. Differentially methylated CpGs in *IFIT1*, *IFI44L*, *MX1*, *RSAD2*, *OAS1*, *EIF2AK2*, *PARP9/DTX3L*, and *RABGAP1L* have been associated with production of autoantibodies[16,21,22]. However, these studies have been performed largely in patients of European descent.

While numerous previous studies focused on either the genetics or epigenetics of SLE, a multi-omics approach coupled with deep clinical phenotyping may better elucidate the molecular basis of disease heterogeneity. By integrating different layers of molecular and clinical data, several studies have provided insight into mechanisms of complex disease such as Alzheimer's disease[23–25], inflammatory bowel disease[26], cancer[27,28], and rheumatoid arthritis[29–32]. In this work, we initially apply unsupervised clustering of ACR classification criteria for SLE to define disease subtypes among a diverse multi-ethnic cohort of SLE patients. We then develop and apply an integrative approach leveraging human genetics and DNA methylation data to elucidate differences between these disease subtypes. We find 256 differentially methylated CpGs that varied significantly according to subtype, of which 61 were under proximal genetic control (Fig. 1).

## Results

**Clinical clustering identifies distinct subtypes of SLE.** Clinical characteristics of the 333 patients examined from the UCSF California Lupus Epidemiology Study (CLUES) cohort are presented in Supplementary Table 1. We first stratified SLE patients into clusters based on ACR classification criteria and sub criteria using an unsupervised clustering approach. Briefly, we first applied multiple correspondence analysis (MCA) and then performed K-means clustering on the top two components chosen by a bootstrap resampling strategy (see Methods). Three clusters were identified. The clusters are labelled M (mild), S1 (severe 1) and S2 (severe 2; Fig. 2a, b). Cluster M was comprised of 101 patients (30.3%) and was characterized by a high prevalence of malar rash, photosensitivity, arthritis, and serositis, but lower prevalence of hematologic manifestations, lupus nephritis, and serologic manifestations ($p < 0.001$). Cluster S1 was comprised of 154 patients (46.2%) and was characterized by higher prevalence of lupus nephritis and anti-dsDNA autoantibody positivity ($p < 0.001$). Cluster S2 was comprised of 78 patients (28.8%) and was the most severe subtype, with a high prevalence of lupus nephritis, autoantibody production (anti-dsDNA, anti-Sm, anti-RNP and antiphospholipid antibodies), and internal organ manifestations, such as hematologic manifestations (Fisher exact test $p < 0.001$; Table 1). The Lupus Severity Index[33], a validated scoring system based on the ACR classification criteria, was also significantly different between the three clusters (ANOVA test $p = 2 \times 10^{-21}$), with cluster M the least severe, and cluster S2 the most severe (Fig. 2c).

With respect to ethnicity, we found a significant increase in the proportion of White patients in cluster M compared to clusters S1 and S2 (Kruskall–Wallis $p = 4.76 \times 10^{-4}$), and a higher proportion of Asian patients in clusters S1 and S2 compared to cluster M (Kruskall–Wallis $p = 1.4 \times 10^{-3}$; Table 1).

At the time of blood sampling, patients in the more severe clusters (S1 and S2) had lower levels of complement C3 (ANOVA $p = 1.47 \times 10^{-3}$), were more likely to be RNP positive (Fisher exact test $p = 3.66 \times 10^{-5}$) and were more likely to be receiving mycophenolate (Fisher exact test $p = 2.48 \times 10^{-3}$) and prednisone (Fisher exact test $p = 5.07 \times 10^{-2}$) than patients in cluster M (Supplementary Table 2). We also examined complete blood counts and proportions taken at time of blood draw from all patients and found a statistically significant decrease in leukocytes (ANOVA $p = 3.31 \times 10^{-3}$), eosinophils (ANOVA $p = 3.39 \times 10^{-2}$) and lymphocytes (ANOVA $p = 2.84 \times 10^{-2}$) among the three clusters (Supplementary Table 2). This could represent a marker of disease severity or a consequence of higher immunosuppressant drug use at the time of blood draw for patients in the more severe disease clusters.

In a comparison of socioeconomic variables across clinical clusters, we did not observe a statistically significant difference in average education level or income between the three clusters (Supplementary Table 3).

**Distinct methylation patterns distinguish clinical clusters.** The clusters identified above, characterized by multiple comorbid phenotypes, represent a clinically relevant framework to stratify SLE patients. Using this stratification, we aimed to identify differentially methylated CpG sites associated with these clinical clusters. Using an ANOVA model, we identified 256 CpG sites in 124 genes that were differentially methylated according to clinical cluster (FDR < 0.1) after adjusting for sex, genetic ancestry principal components, cell composition, medications, alcohol use, and smoking status (Fig. 3a; Supplementary Data 1). A quantile-quantile plot is shown in Supplementary Figure 1. The observed versus expected test statistic demonstrates no evidence for inflation of the association tests (inflation factor $\lambda = 0.99$).

Upon mapping these 256 cluster-associated CpG sites to genes and performing pathway analysis, we found significant enrichment of genes associated with Type I interferon signaling, antiviral responses and inflammatory pathways (gene list enrichment analysis FDR < 0.01; Table 2).

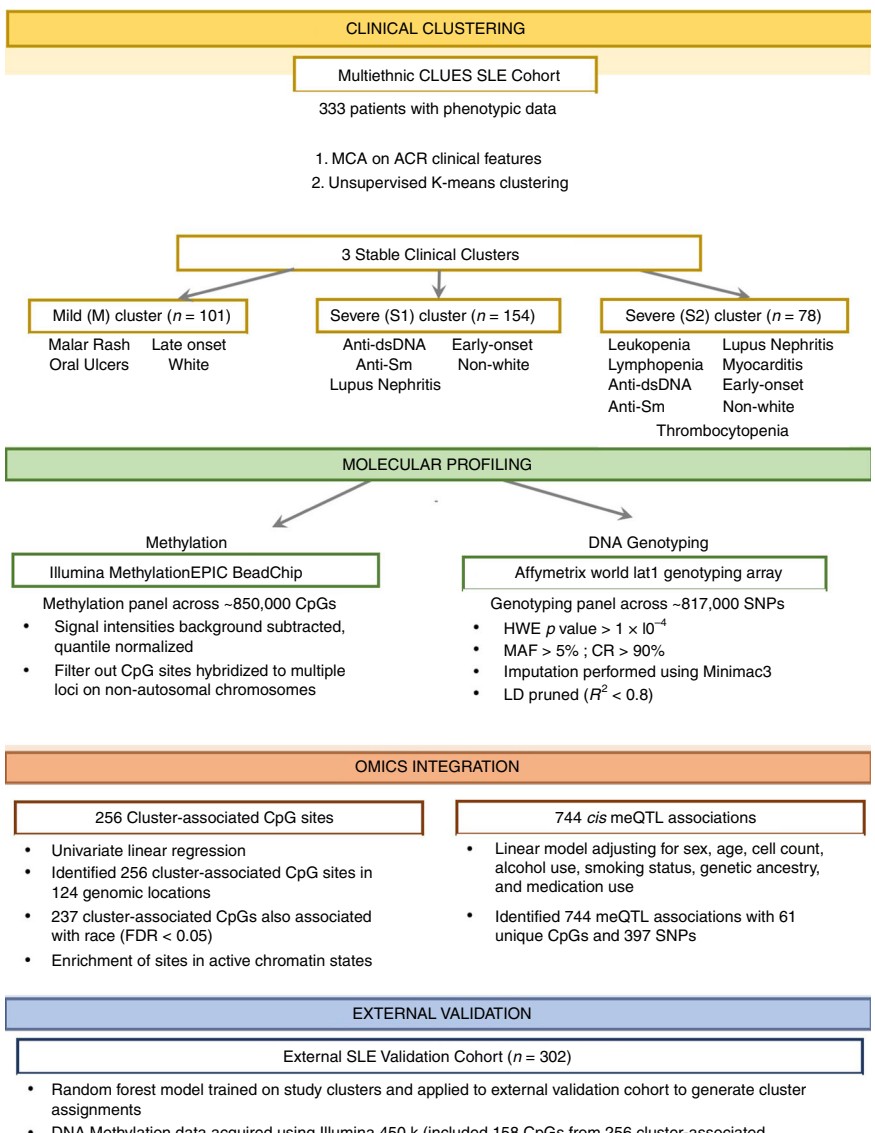

**Fig. 1** Integrative analysis pipeline. An overview of the omics data integration strategy used to characterize clinical clusters identified by K-means clustering. MCA = Multiple Component Analysis, HWE = Hardy-Weinberg Equilibrium, MAF = minor allele frequency, LD = linkage disequilibrium, FDR = false discovery rate, meQTL = cis-methylation quantitative trait loci

In order to functionally classify the cluster-associated CpGs, we intersected these genomic regions with the Hallmark Interferon-Alpha Responsive gene set[34] since the IFN-alpha signaling pathway has been previously implicated in SLE pathology[18,35–38]. We observed a significant enrichment of IFN-alpha responsive genes (hypergeometric $p < 0.01$) with 101 out of the 256 CpGs within this set. Notably, of the 101 IFN-alpha CpGs, 93 were hypermethylated in cluster M relative to both cluster S1 and S2. Of these CpGs, 57 were in the promoter region (TSS200, TSS1500, 5′ UTR), and 36 were in the gene body. Hypermethylation at the promoter sites suggests a role for epigenetic silencing in cluster M with respect to S1 and S2 while gene body hypermethylation suggests gene expression.

Cluster-associated CpGs with the greatest variance (5–11% methylation variance) across the clusters were in genes *IFI44L, MX1, PARP9, EPSTI1,* and *PDE7A*, all displaying hypermethylation in cluster M relative to S1 and S2 (Supplementary Data 2). With the exception of *PDE7A*, all of these genes are interferon responsive. *PDE7A* encodes a phosphodiesterase associated with

T cell activation and IL-2 production[39]. Differentially methylated CpGs in *IFI44L, MX1,* and *PARP9* map to the 5-UTR region, suggesting silencing of these genes. Differentially methylated CpGs in *EPSTI1* and *PDE7A* are located in the gene body, where hypermethylation is associated with gene expression.

For each of the 256 CpGs identified above using the ANOVA test, we then sought to determine which pairwise comparison (cluster S2 vs M, S2 vs S1, or S1 vs M) contributed to the significant F-statistic. Using the nestedF method in the Limma R package[40], 247 of the aforementioned associated 256 CpGs were differentially methylated between clusters S2 and M (FDR $p < 0.1$; Supplementary Fig. 2A, Table 3 and Supplementary Table 5). The most significant CpGs were in the promotor of *IFI44L* and gene body of *RSAD2*, with hypermethylation in cluster S1 versus M. Comparison of clusters S2 to S1 identified 18 differentially methylated CpGs (FDR < 0.1; Supplementary Fig 2B, Table 3 and Supplementary Table 5), with hypermethylation of CpGs in *IFI27* and *B2M*, a component of the MHC1 complex. Comparison of clusters S1 and M identified 53 differentially methylated CpGs

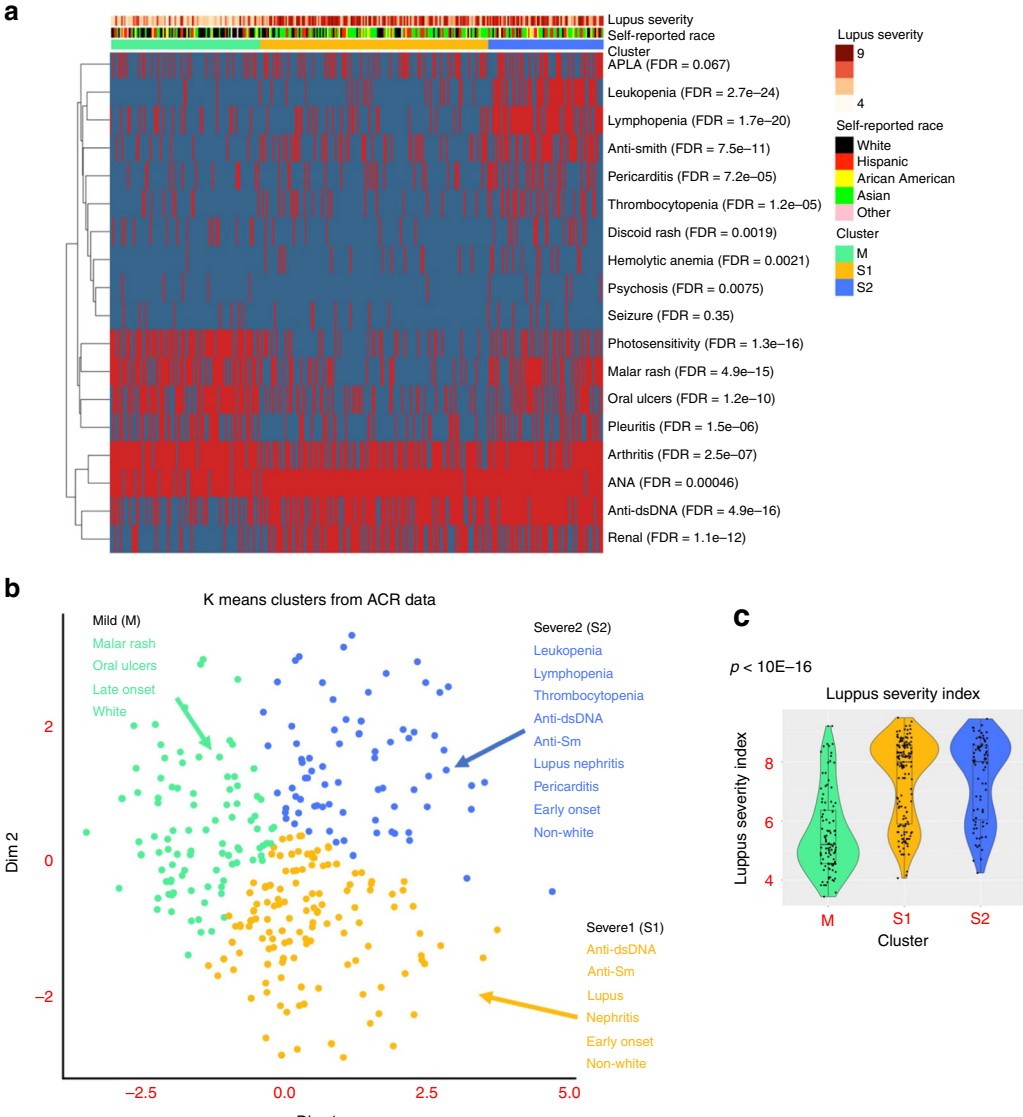

**Fig. 2** Characterization of clinical features between clusters. **a** Distribution of American College of rheumatology (ACR) classification criteria for SLE within each cluster where red indicates presence and blue absence of each criterion. Association between each criterion and cluster was evaluated by a Fisher exact test. **b** Criteria significantly associated with each cluster (FDR < 0.01). **c** Distribution of lupus severity index across clusters with p-value computed using an ANOVA test

(FDR < 0.1; Supplementary Fig. 2C; Table 3 and Supplementary Table 5). The percent variance between clinical clusters explained by CpG methylation varied from 0.9% for cg23002431 (COPA gene body) to 21% for cg00959259 (PARP 5'UTR).

**Validation of cluster-specific methylation profiles**. To determine whether the methylation signature associated with the three clinical clusters identified was reproducible, we applied our clustering method to a previously published independent cohort of 302 female SLE patients of European descent[16]. This cohort has a lower lupus severity index (6.15 ± 1.42) compared to the CLUES cohort (6.85 ± 1.63; Student's t test p < 0.001), however this difference is relatively small and not clinically significant. In order to identify methylation associations in this validation cohort, we first assigned a cluster label (M, S1, or S2) to each patient using the study cohort as a reference. Since the clusters in the study cohort were discovered using an unsupervised approach, we first trained a random forest model on the CLUES data with the ACR features as input. Model parameters were optimized by minimizing the

out-of-bag error (Supplementar Fig. 3). The model achieved a minimum out-of-bag error of 12.8%. We then applied this model to the validation cohort of SLE patients of European descent and determined a cluster label for each sample. Clinical characteristics of subjects in each cluster in the validation cohort are presented in Supplementary Table 4. In comparison to the CLUES cohort where the majority of subjects were in cluster S1, the majority in the validation cohort were in cluster M, reflecting the racial differences between the cohorts.

To determine whether the methylation patterns associated with each cluster were robust and reproducible, we evaluated methylation differences in the independent cohort of 302 female SLE patients of European descent as described above[16]. Since the validation dataset was obtained using the Illumina 450k BeadChip, we restricted these analyses to the 158 cluster-associated CpGs in the CLUES cohort that were also on the 450k array. Of these 158 CpGs, 132 (84%) were significantly associated with cluster in the validation dataset (FDR < 0.1; Table 3). We observed a strong correlation (r > 0.9) between differences in methylation beta values for the CLUES

**Table 1 Summary of significant clinical and demographic variables across clusters**

| | Cluster | | | P value | FDR |
|---|---|---|---|---|---|
| | **M (N = 101)** | **S1 (N = 154)** | **S2 (N = 78)** | | |
| *ACR criteria (%)* | | | | | |
| Malar rash | 68.3 | 20.8 | 62.8 | 1.22E-15 | 4.87E-15 |
| Discoid rash | 14.9 | 5.2 | 20.5 | 1.41E-03 | 1.88E-03 |
| Oral ulcers | 70.3 | 26.6 | 48.7 | 4.72E-11 | 1.18E-10 |
| Arthritis | 93.1 | 66.2 | 88.5 | 1.14E-07 | 2.53E-07 |
| Pleuritis | 48.0 | 17.5 | 38.5 | 7.41E-07 | 1.48E-06 |
| Pericarditis | 13.0 | 13.6 | 35.9 | 4.71E-05 | 7.24E-05 |
| Seizure | 8.0 | 4.5 | 9.0 | 0.354 | 0.354 |
| Psychosis | 5.0 | 0.6 | 9.0 | 6.79E-03 | 7.54E-03 |
| anti-dsDNA antibodies | 35.6 | 76.0 | 92.3 | 9.79E-17 | 4.90E-16 |
| anti-Smith antibodies | 9.9 | 27.9 | 57.7 | 2.64E-11 | 7.55E-11 |
| ANA positivity | 89.1 | 98.7 | 98.7 | 3.20E-04 | 4.57E-04 |
| Hemolytic anemia | 1.0 | 8.5 | 15.4 | 1.65E-03 | 2.07E-03 |
| Leukopenia | 4.0 | 11.7 | 65.4 | 1.34E-25 | 2.68E-24 |
| Lymphopenia | 19.8 | 16.9 | 76.9 | 1.67E-21 | 1.67E-20 |
| Thrombocytopenia | 8.9 | 13.0 | 34.6 | 6.82E-06 | 1.24E-05 |
| Renal | 15.8 | 62.3 | 57.7 | 3.32E-13 | 1.11E-12 |
| Photosensitivity | 73.3 | 18.2 | 41.0 | 1.91E-17 | 1.27E-16 |
| APLA | 25.7 | 32.5 | 42.3 | 0.0641 | 0.0675 |
| *Ethnicity* | | | | | |
| White | 48.0 | 22.7 | 16.7 | 4.76E-04 | 1.90E-03 |
| Hispanic | 18.0 | 22.1 | 30.8 | 0.115 | 0.23 |
| African–American | 11.0 | 9.7 | 12.8 | 0.642 | 0.64 |
| Asian | 20.0 | 44.8 | 37.2 | 1.43E-03 | 4.30E-03 |
| Other | 3.0 | 0.6 | 2.6 | 2.57E-05 | 1.29E-04 |
| Lupus Severity Index (SD) | 5.6 (1.37) | 7.39 (1.43) | 7.41 (1.42) | 2.93E-22 | 2.05E-21 |
| SLEDAI Score (SD) | 2.43 (2.94) | 2.82 (2.9) | 3.94 (3.44) | 3.58E-03 | 1.08E-02 |

*ACR* American college of Rheumatology, *APL* antiphospholipid antibodies, *FDR* false discovery rate, *SLEDAI* SLE disease activity index
False Discovery Rate (FDR) p-values were calculated for Kruskall–Wallis (continuous variables) or Fisher's exact test (binary variables)

cohort and validation set for all three pairwise comparisons (cluster S1 vs. M, S2 vs. S1, and S2 vs. M; Table 3, Fig. 4).

**Active chromatin states in cluster-associated CpGs.** In order to further characterize the epigenetic landscape of the cluster-associated CpGs, we examined CpG enrichment in genomic regions classified according to specific chromatin states based on the Epigenome Roadmap 15 state model[41]. Results for 13 peripheral blood cell types are summarized in Supplementary Fig 4. We found significant depletion (Fisher's exact FDR < 0.01; OR < 0.5) of cluster-associated CpGs in quiescent regions in 12 of the 13 cell types, and significant enrichment (FDR < 0.01; OR > 2) in enhancers and regions flanking active transcriptional start sites in all cell types. We also observed significant enrichment (Fisher's exact FDR < 0.01; OR > 2) of H3K4me3, H3K4me1, and H3K27ac histone marks specific for active enhancers in all peripheral blood cell types (Supplementary Fig 5).

Epigenetic annotation of differentially methylated CpGs in *IFI44L* land in enhancers and active transcription sites in peripheral blood primary B cells, T helper memory cells, Naïve T cells, Th17 cells, T memory cells and T regs, but not in regulatory or transcription sites in neutrophils or NK cells. Differentially methylated CpGs in *MX1, PARP9, EPSTI1,* and *PDE7A* are located in enhancers and transcription sites in most peripheral immune cell subtypes.

**meQTL loci controlling cluster-associated CpGs.** We sought to understand the sources of methylation differences in the clinically-defined clusters. Therefore, we used paired genotype data to investigate genetic drivers of methylation differences. Specifically, we conducted a methylation quantitative trait loci

analysis (meQTL) to determine which cluster-associated CpGs were under proximal genetic control (distance between SNP and CpG < 1 Mb). Genetic data was first imputed and LD-pruned ($r^2 < 0.8$). After adjusting for population structure, sex, age, cell type composition, medication use, smoking status, and alcohol consumption, we found 744 significant *cis* meQTL associations (FDR < 0.05; Fig. 3b). These involved 61 unique CpGs in 41 genes, and 397 SNPs in 90 genes (Supplementary Data 3).

Of the 744 significant *cis* meQTL associations, 91 meQTLs in 19 unique CpGs were in interferon-alpha or interferon-gamma responsive genes. Of these, the greatest number of meQTL loci were in *EPSTI1* (12 SNPs), *PARP14* (nine SNPs), and *PARP15* (8 SNPs) for CpGs in interferon-alpha responsive genes (Supplementary Data 3). We found 39 meQTL associations involving CpG sites in the promoter region. Notably, we found 21 associations in CpGs in *PARP14*, of which 20 were in the promoter region under the control of SNPs in *PARP14, PARP15,* and *DIRC2*. We also found CpGs in the promoter region of OAS3 (*n* = 2) under the control of SNPs in *LHX5-AS1*, and one CpG in *USP18* under the genetic control of SNPs in *LINC01634*.

Of the non-interferon-responsive CpGs, we found 20 genetic variants that controlled methylation of cg07259759 located in the gene body of *USP35*, a ubiquitin specific peptidase[42] (methylation variance 21–25%). Ten of these 20 genetic variants were found in an intron of *GAB2*, a tyrosine kinase adaptor that is primarily upregulated in activated innate immune cells[43–45]. We also found 43 genetic variants in *HLA-F*, a MHC-Ib minor allele involved in NK cell self-recognition[46], which controlled methylation at four CpG sites in the gene body of *HLA-F*. Fifteen CpGs were located in the promoter or 5'UTR region, with the largest methylation variance observed for cg04738877 in the promoter

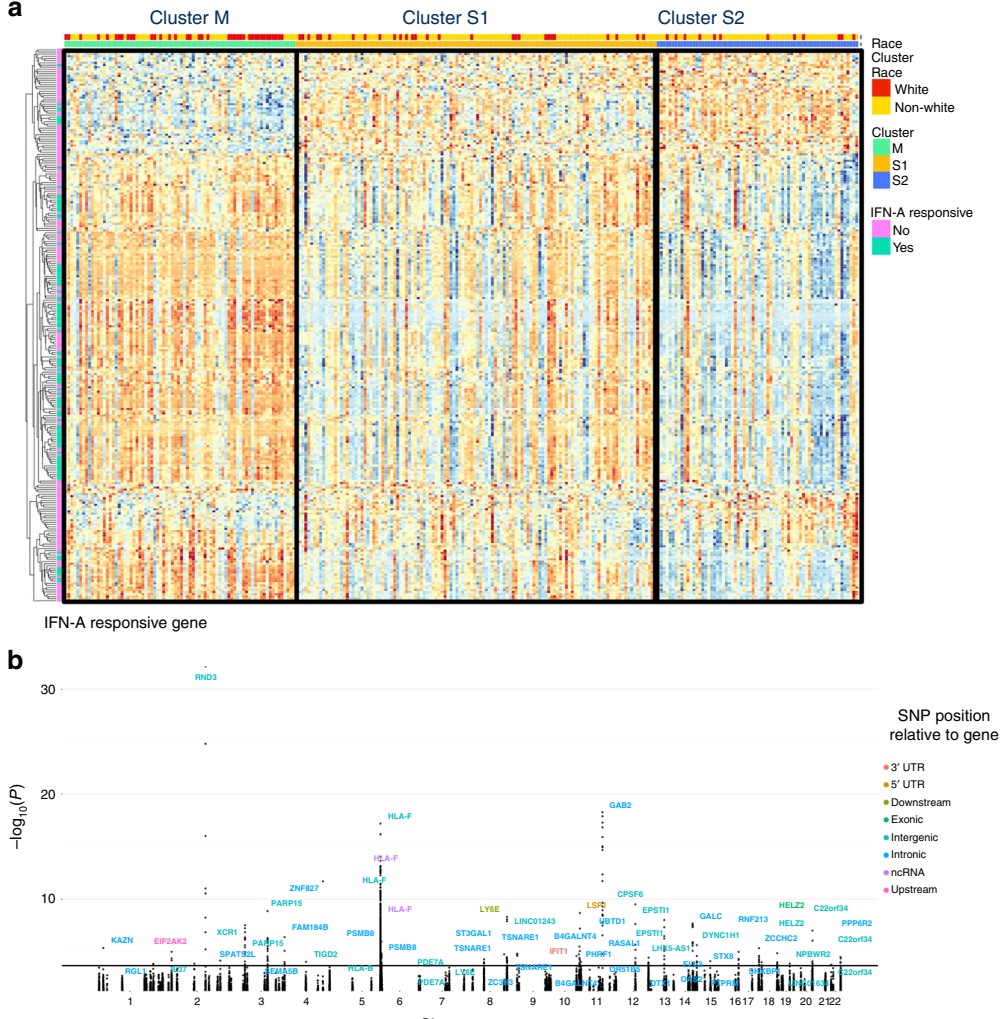

**Fig. 3** Cluster associated CpGs and meQTL associations. **a** Heatmap of CpGs significantly associated with clinical cluster (FDR < 0.1) **b** Manhattan plot shows −log10(p-value) for associations between cluster-associated CpGs and all SNPs within 1 Mb of each CpG. For each CpG with a significant meQTL (FDR < 0.05), the most significant variant is labelled with its corresponding gene

region of *GALC*, under the control of SNPs in introns of the same gene.

Although we considered all SNPs within a 1 Mb window around each CpG, the proportion of significant meQTL associations decreased as the distance between SNPs and CpGs increased (Supplementary Fig. 6A and B). This suggests that genetically determined CpG methylation was typically driven by proximal genetic variation, rather than distal effects.

**Epigenetic mediation of genetic association with clusters**. One challenge of interpreting the methylation associations with clusters is that many methylation differences may represent a consequence of clinical differences between clusters rather than causal mediators. In order to identify which CpGs may mediate genetic associations with clusters and reveal novel biology, we employed an integrative causal inference method[47].

Briefly, this method uses conditional probabilities to evaluate a causal relationship between a factor (genotype), a potential mediator (CpG methylation), and an outcome (clinical cluster).

First, from the list of 744 meQTLs identified, we selected the SNPs that were significantly associated with cluster (FDR < 0.05). For these meQTL associations, we identified the subset where methylation appears to mediate the genotype-cluster association

using the causal inference test (CIT). This yielded 24 meQTLs with 21 SNPs (FDR < 0.05; Supplementary Table 5). Notably from these, we found 6 significant associations between SNPs in *GAB2* and CpGs in *USP35*. We also found evidence for methylation mediation of SNPs in *HLA-F*. Figure 5 provides an example of one of these associations between a SNP in *GAB2* and CpG in *USP35*.

**Ethnicity-associated differentially methylated CpGs**. As some of the methylation differences in the clinically-defined clusters could be explained by genetic variation in the meQTL analysis, we explored the effect of ethnicity, after adjusting for genetic factors. Previous work has identified patterns of differential methylation across ethnic groups due to both ancestral genetic variation and environmental influences[48]. As non-White ethnicity is associated with worse outcomes in SLE, we sought to determine whether the differentially methylated CpGs across clusters were enriched for ethnicity-associated CpGs, after adjusting for genetic ancestry. Of the 256 cluster-associated CpGs, we identified 237 CpGs that were associated with ethnicity (FDR < 0.05) after adjusting for sex, the top three genetic ancestry principal components, cell composition, medications, alcohol use, and smoking status. A permutation analysis was conducted by randomly permuting

**Table 2 Significantly enriched pathways in cluster-associated CpGs**

| Name | Source | P value | FDR B&H | Genes from Input | Genes in Annotation |
|---|---|---|---|---|---|
| Interferon signaling | REACTOME | 9.54E-32 | 8.19E-29 | 29 | 202 |
| Interferon alpha/beta signaling | REACTOME | 2.06E-30 | 8.84E-28 | 21 | 69 |
| Cytokine signaling in immune system | REACTOME | 2.26E-20 | 6.46E-18 | 34 | 763 |
| Interferon gamma signaling | REACTOME | 1.04E-15 | 2.23E-13 | 14 | 94 |
| Influenza A | KEGG | 3.35E-13 | 5.76E-11 | 15 | 173 |
| Herpes simplex infection | KEGG | 8.98E-13 | 1.29E-10 | 15 | 185 |
| Measles | KEGG | 1.09E-09 | 1.33E-07 | 11 | 134 |
| ISG15 antiviral mechanism | REACTOME | 1.67E-09 | 1.59E-07 | 9 | 77 |
| Antiviral mechanism by IFN-stimulated genes | REACTOME | 1.67E-09 | 1.59E-07 | 9 | 77 |
| Hepatitis C | KEGG | 1.33E-08 | 1.14E-06 | 10 | 131 |
| RIG-I/MDA5 mediated induction of IFN-alpha/beta pathways | REACTOME | 7.27E-08 | 5.68E-06 | 8 | 84 |
| Negative regulators of RIG-I/MDA5 signaling | REACTOME | 1.13E-07 | 8.08E-06 | 6 | 36 |
| TRAF3-dependent IRF activation pathway | REACTOME | 3.00E-06 | 1.87E-04 | 4 | 16 |
| TRAF6 mediated IRF7 activation | REACTOME | 3.05E-06 | 1.87E-04 | 5 | 35 |
| RIG-I-like receptor signaling pathway | KEGG | 6.35E-06 | 3.64E-04 | 6 | 70 |
| Antigen presentation: folding, assembly and peptide loading of class I MHC | REACTOME | 1.99E-05 | 1.01E-03 | 4 | 25 |
| TRAF6 mediated NF-kB activation | REACTOME | 1.99E-05 | 1.01E-03 | 4 | 25 |
| Viral carcinogenesis | KEGG | 5.15E-05 | 2.46E-03 | 8 | 201 |
| Endosomal/vacuolar pathway | REACTOME | 5.81E-05 | 2.62E-03 | 3 | 12 |
| NF-kB activation through FADD/RIP-1 pathway mediated by caspase-8 and -10 | REACTOME | 7.51E-05 | 3.23E-03 | 3 | 13 |
| NOD-like receptor signaling pathway | KEGG | 1.26E-04 | 5.14E-03 | 7 | 170 |
| Nicotinamide salvaging | REACTOME | 2.09E-04 | 8.17E-03 | 3 | 18 |
| ER-Phagosome pathway | REACTOME | 2.65E-04 | 9.91E-03 | 5 | 87 |

FDR B&H false discovery rate by Benjamini and Hochberg method
CpGs were mapped to genes using Illumina annotation file, and pathway analysis was performed using ToppFunn[92]

**Table 3 Summary of cluster-wise comparison and validation**

| Comparison | Number of Differentially Methylated CpGs in study cohort (# CpGs on 450k Chip) | Significant in Validation Set (FDR < 0.1) | ΔBeta R-squared CLUES vs. Validation |
|---|---|---|---|
| Cluster S1 vs M | 53 (28) | 21 | 0.94 |
| Cluster S2 vs S1 | 18 (8) | 6 | 0.96 |
| Cluster S2 vs M | 247 (122) | 105 | 0.94 |

Rows indicate individual pairwise comparisons as performed using the nestedF method in Limma

ethnicity 1000 times and testing for association with ethnicity. Figure 6a displays the density of ethnicity-associated CpGs in 1000 random samples. This analysis revealed a significant ($p < 0.001$) enrichment of ethnicity-associated CpGs in the cluster-associated methylation signature. Results and data are available as an RShiny Application for use in future research: http://comphealth.ucsf.edu/sle_clustering/.

**Discussion**

In the present study, we developed a stepwise multi-omics approach for identifying SLE patient subtypes defined by clinically-relevant phenotypes and molecular mechanisms among a multi-ethnic cohort. We report three lupus clinical subtypes defined by the ACR classification criteria that vary according to disease severity. We also show that patterns of differential methylation at specific CpGs are associated with the clinical subtypes. A subset of these CpGs are under genetic control, however the majority display a strong association with ethnicity after adjusting for genetic ancestry, suggesting possible molecular mediators of the ethnic-effect underlying lupus outcomes.

Unlike previous studies that have largely studied SLE patients of European genetic ancestry, we studied a cohort or patients of White, African-American, Asian, and Hispanic ethnicity. Since

SLE severity is known to vary widely between racial and ethnic groups, analysis of a large multiethnic cohort is crucial for understanding the genetic and non-genetic determinants of this ethnic-associated variability.

Unsupervised clustering approaches have been applied widely to high dimensional omics datasets with the aim of deriving meaningful clusters characterized by a small set of molecular features[6,49,50]. By translating this dimensionality reduction technique to the ACR clinical features in SLE, we identified three clinical subtypes each characterized by specific ACR features. Due to the strong association of DNA methylation with genetic variation, unsupervised clustering of the methylation data revealed population structure rather than lupus-relevant clinical differences. For these reasons, we did not report DNA methylation clustering and rather chose to define subtypes by ACR criteria. Since this is an ongoing cohort, with a larger sample size, we may be able to define methylation-based clusters in each racial group separately, minimizing the effect of genetic structure confounding.

Importantly, the clusters defined in this study are consistent with previous epidemiological studies describing the correlation of multiple sub-phenotypes of SLE, such as the correlation of SLE skin manifestations with arthritis, serositis with the lack of

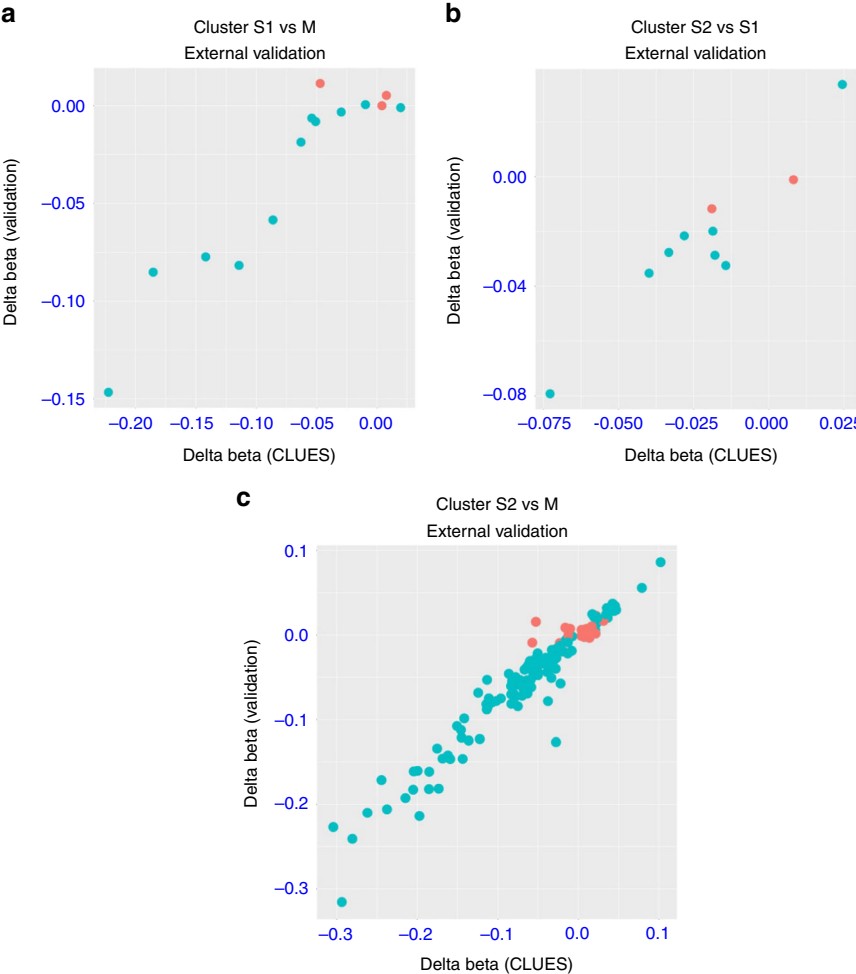

**Fig. 4** Validation of differentially methylated CpGs between clusters. Differentially methylated CpGs between clusters were validated in an external dataset (19) **a** cluster S1 vs M. **b** cluster S2 vs S1 **c** cluster S2 vs M. Difference in CpG beta values for differentially methylated sites in CLUES data on x axis and corresponding delta beta value for CpG in validation data on y axis. Green color indicates significant association with cluster in validation dataset (FDR < 0.1)

other end-organ involvement, and anti-dsDNA with lupus nephritis[21,51,52]. The milder subtype in this study had a higher prevalence of participants of White race. This has also been previously described, as patients with European ancestry have a higher proportion of arthritis, skin manifestations and serositis and lower prevalence of lupus nephritis and autoantibody production[53–55].

By considering clusters defined by multiple phenotypes we preserve the multifactorial clinical nature of SLE. Training a random forest model using the cluster assignments as labels allowed us to apply this clustering scheme to an external dataset of patients of European descent. After validation in other larger multi-ethnic cohort, this model may form a clinically useful stratification method to further study disease heterogeneity in SLE.

After identifying clinically-relevant patient clusters, we found a set of 256 CpGs associated with the clusters with strong enrichment of methylation for genes in the Type I interferon pathway, cytoplasmic viral sensing pathways, and immune related pathways, with significant enrichment in enhancers and regions flanking active transcriptional start sites in all peripheral immune cell types. Several studies have implicated transcriptional upregulation and epigenetic regulation of the Type I interferon pathway in SLE[17,19,35–37,56]. We and others have also previously described methylation changes in interferon

responsive genes associated with individual lupus outcomes, including cutaneous rash[57], renal involvement[19,20,58] and serologic manifestations[16,22]. However, given the striking heterogeneity of clinical features in SLE, epigenetic programs associated with single phenotypes may be less relevant in a clinical setting. By performing unsupervised clustering on a diverse SLE cohort, we can study the molecular heterogeneity in clinically-relevant subtypes driven by multiple SLE outcomes and disease severity. With this approach we found that severe subtypes, which also have higher proportions of patients of Asian and Hispanic ethnicity, have a higher degree of type I interferon dysregulation. Of these CpG sites, the greatest methylation variance across the clusters was in IFI44L, which encodes for Interferon Induced Protein 44 Like, with progressive hypomethylation from cluster M to cluster S2. Although the function of IFI44L is unknown, increased IFI44L expression is a component of the type-1 IFN response signature and also part of the cellular response to viral infections[59]. IFI44L promoter methylation has been proposed as a blood biomarker for SLE[58].

Since genetic variation can have profound effects on DNA methylation[60–62], we also performed an meQTL analysis to quantify the degree of proximal genetic control of the 256 CpG signature. Although the cluster-associated CpG set was strongly enriched for Type 1 IFN genes, only a subset of these Type 1 IFN CpGs (24%) had meQTL loci, suggesting that environmental

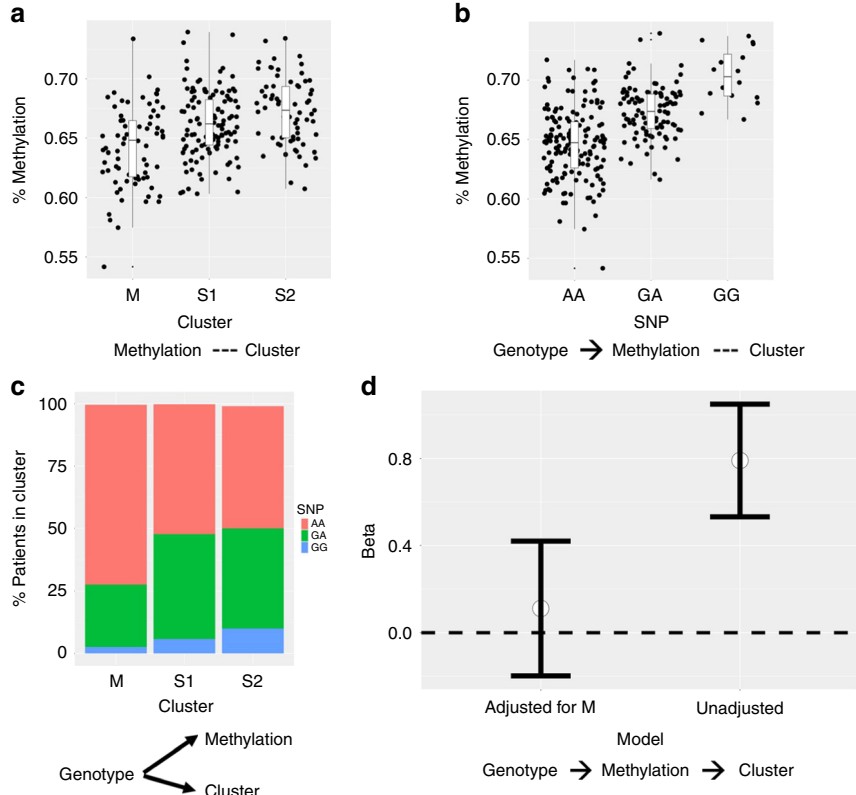

**Fig. 5** Identification of candidate cluster-associated cg07259759 (*USP35*) that mediates genetic association of rs7104222 (*GAB2*) with clusters. Association of DNA methylation of cg07259759 and cluster (**a**) or genotype of rs7104222 (**b**). **c** Association between genotype rs7104222 and clusters. **d** Beta coefficient represents the dependence of cluster on genotype with or without adjusting for methylation. Error bars represent the 95% confidence interval for beta coefficient estimate. After adjusting for methylation, the observed dependence reduces toward zero

factors may be contributing more to the epigenetic regulation of the Type 1 IFN pathway than genetic factors. We found interesting associations between genetic variation and methylation at CpGs in immune relevant genes, and in 24 meQTL associations we had evidence of mediation of the genetic association by methylation at the corresponding CpG. We would like to highlight variants in *HLA-F, PAR14* and *GAB2* controlled methylation sites in *USP35. HLA-F* is part of the nonclassical *HLA-Ib* genes, which are mono- or oligomorphic[46]. Surface expression of *HLA-F* has been demonstrated on activated T, B and NK cells, and serum IgG autoantibodies against *HLA-F* have been detected in SLE patients and correlated with disease activity[63–65]. *PARP14* encodes for poly(ADP-ribose) polymerase (PARP) protein family 14 and is involved in cellular maintenance and cell fate decisions, such as cell-cycle progression, metabolic pathways and ribosome biogenesis[66]. Its role in SLE and autoimmune disease has not been defined but it has been shown to regulate glycolysis via IL-4 in B lymphocytes[67] and to promote survival of cancer cells[67–69]. Glycolysis in SLE has been found to directly influence the Th17 cell fate and survival, therefore implicating a potential mechanistic role for PARP14 in SLE[70].

*GAB2* is a member of the GRB2-associated binding protein (GAB) gene family. These genes act as adapters for transmitting various signals in response to stimuli through cytokine and growth factor receptors, and T- and B-cell antigen receptors[45]. Among its related pathways is Akt signaling, which is involved in cell proliferation and autophagy, a process that has been implicated in SLE pathogenesis[44,71–73]. Variants of *GAB2* influenced methylation marks in the gene body of *USP35*, which encodes for a member of the peptidase C19 family of ubiquitin-specific proteases[42]. This deubiquitinating enzyme has been shown to

mediate the IFN-type I response upon viral infection and it has been associated with higher IFN-β and IFIT1 gene expression[74]. This is relevant to our findings as higher levels of IFN-β have been associated with SLE[75–77]. As with all epidemiologic studies, these results represent hypotheses that will require mechanistic validation and independent replication.

Variation in methylation can be attributed to genetic and non-genetic effects. The majority of differentially methylated CpGs among disease subtypes were not classified as under genetic control. Although the number of detected meQTL associations is likely to increase with a larger sample size, it also suggests a greater role for non-genetic or environmental effects. Since self-reported race and ethnicity refers to communality in cultural heritage, language, social practice, traditions, and geopolitical factors, it may be a proxy for a wide variety of environmental exposures not easily captured. Since ethnicity plays a role in differences in lupus severity, we were interested in the association of ethnicity and our methylation findings, after adjusting for genetic variation. After adjusting for genetic principal components, we found a significant enrichment of self-reported race-associated CpGs in our 256 CpG signature ($p < 0.001$). This suggests that there may be a common set of CpGs that mediates both SLE clusters and non-genetic differences in race. Although this might raise concern for confounding by genetic structure, our analysis revealed a low genomic inflation factor. Furthermore, the CpGs are biologically relevant in SLE pathogenesis. In addition, previous work has identified differential methylation between ethnic groups due to environmental factors that is not fully explained by genetic ancestry[48]. It is well known that SLE outcomes vary according to race, and the causes behind these race disparities are a source of ongoing debate. We observed

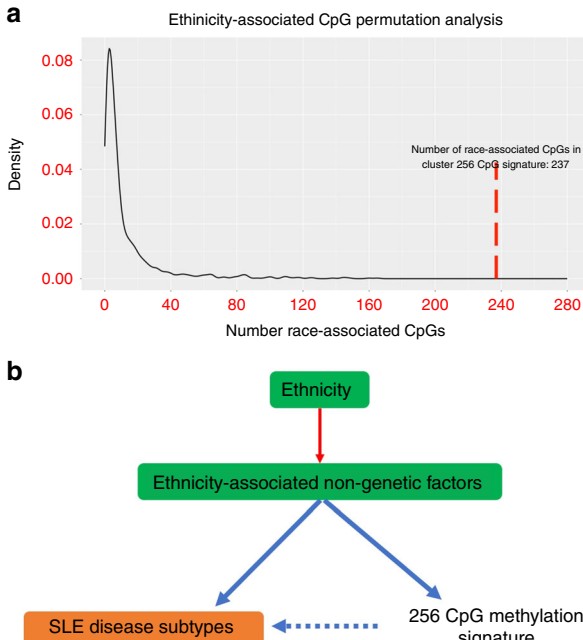

**Fig. 6** Enrichment of ethnicity-associated CpGs in set of cluster-associated CpGs. **a** Null distribution generated by randomly permuting ethnicity labels 1000 times and identifying the number of cluster associated CpGs that were also significantly with ethnicity (*p* < 0.05) in each sample. The red line indicates the number of significant ethnic-associated CpGs (238) found in the set of 256 cluster-associated CpGs. **b** Working model illustrating the role of ethnic-associated non-genetic factors in controlling both SLE disease subtypes and methylation signature

differences in race across the disease subtypes, with the milder subtype having a larger representation of White patients. Therefore, these methylation differences suggests the existence of molecular mediators of the non-genetic race-related clinical differences in SLE outcomes, and may reflect environmental exposures that affect races differently. Figure 6b displays a model for the role of race-associated non-genetic factors that control both methylation of the 256 CpG signature and SLE disease subtypes.

We applied this approach to an external cohort of exclusively SLE patients of European descent. The clinical clusters were labelled by applying a random forest model trained on the original CLUES clusters. The greatest number of subjects were assigned to cluster M. Since cluster M was also enriched for patients that self-identified as White in the original clusters, we believe that the clustering model accurately captures the role of racial differences in lupus severity. We were only able to test 158 of the 256 cluster-associated CpGs in the validation cohort due to a limited number of overlapping probes. However, we validated 84% of the cluster-associated methylation sites. As we also found that the majority of cluster-associated CpGs were race-associated in the CLUES cohort, this suggests that while race plays a role in DNA methylation, these effects are secondary to the clinical differences between lupus clusters. As we could not determine the relative genetic and non-genetic contributions of the methylation differences, we cannot recreate the race-specific findings we have described for the CLUES cohort. In general, this suggests that the subtypes and associated methylation differences we have described can be applied to other cohorts.

Our study has likely been limited by the modest sample size, in particular the small number of African-American SLE patients. We have also profiled DNA methylation marks in whole blood, which even after rigorous adjustment methods may be subject to

confounding due to cell composition. Additionally, since methylation changes may be cell-type specific, our data may be insensitive to these effects. For example, when comparing clusters S1 and S2, we were only able to detect 20 differentially methylated CpGs. However, since clusters S1 and S2 had significantly different rates of lymphopenia and leukopenia, methylation differences may represent both differences in cell proportion and cell-type specific epigenetic remodeling that cannot be distinguished in whole blood data. Additionally, our validation cohort is limited not only by its ethnic and gender composition, but also since methylation was measured on a different chip than the present study. This meant that we were only able to test a subset (158) of the 256 cluster-associated CpGs in the validation cohort. Ideally, clustering would be performed separately in each racial group to account for possible race-specific effects. However, due to the limited sample size, this was not feasible in the current study. Future work with larger sample sizes in non-European populations may help to address this issue.

Strengths of this study include the rich phenotyping data and adjustments for major confounders, including medications at the time of blood draw, smoking history, and alcohol consumption, which are unaccounted for in most epigenome-wide association studies. This is also the largest cohort including African American, Caucasian, Asian, and Hispanic patients to be profiled for genome wide DNA methylation and genotyping, which allowed us to differentiate between genetic and non-genetic effects of race in SLE outcomes, shedding light on molecular mediators of race in disease heterogeneity. Future work will include testing these findings in other multi-ethnic cohorts. Furthermore, it will be of interest to determine whether these DNA methylation differences are predictive of future disease activity and severity.

In summary, we have identified three distinct clinical subtypes of SLE that have distinct patterns of methylation at specific CpG sites. While previous studies have defined subtypes based on transcriptomic data[6–8], by integrating methylation and genetic data, the three subtypes identified here may reflect the influence of both genetic and non-genetic effects. We also identified potential mediation of genetic association by methylation changes not previously addressed in SLE. Furthermore, we have demonstrated the utility of studying a diverse SLE population to investigate the molecular underpinnings of race differences in SLE outcomes.

## Methods

**Subjects and samples.** Subjects were participants in the California Lupus Epidemiology Study (CLUES), a multi-racial/ethnic cohort of individuals with physician-confirmed SLE. This study was approved by the Institutional Review Board of the University of California, San Francisco. All participants signed a written informed consent to participate in the study. Participants were recruited from the California Lupus Surveillance Project, a population-based cohort of individuals with SLE living in San Francisco County from 2007 to 2009 (2, 47). Additional participants residing in the geographic region were recruited through local academic and community rheumatology clinics and through existing local research cohorts.

Study procedures involved an in-person research clinic visit, which included collection and review of medical records prior to the visit; a history and physical examination conducted by a physician specializing in lupus; collection of biospecimens, including peripheral blood for clinical and research purposes; and completion of a structured interview administered by an experienced research assistant. All SLE diagnoses were confirmed by study physicians based upon one of the following definitions: (a) meeting ≥ 4 of the 11 American College of Rheumatology (ACR) revised criteria for the classification of SLE as defined in 1982 and updated in 1997[5,78], (b) meeting 3 of the 11 ACR criteria plus a documented rheumatologist's diagnosis of SLE, or (c) a confirmed diagnosis of lupus nephritis, defined as fulfilling the ACR renal classification criterion (>0.5 grams of proteinuria per day or 3 + protein on urine dipstick analysis) or having evidence of lupus nephritis on kidney biopsy.

**DNA methylation assessment.** DNA methylation of genomic DNA from peripheral blood mononuclear cells was profiled using the Illumina MethylationEPIC

BeadChip. This chip assesses the methylation level of ~850,000 CpGs in enhancer regions, gene bodies, promoters, and CpG islands. All subsequent processing was done using the R minfi package. Signal intensities were background subtracted using the minfi noob function and then quantile normalized[79,80]. Sites with a poor detection rate (detection $p$ value > 0.05) in more than 5% of the samples (1746 CpG sites) were removed. Sites predicted to hybridize to multiple loci (44,097 CpG sites) and sites on non-autosomal chromosomes (19,627 CpG sites) were removed, resulting in 802,579 probes for analyses.

**DNA genotyping.** Genotyping for genomic DNA from peripheral blood mononuclear cells was performed using the Affymetrix Axiom Genome-Wide LAT 1 Array. This genotyping array is composed of 817,810 SNP markers across the genome and was specifically designed to provide maximal coverage for diverse racial/ethnic populations, including West Africans, Europeans and Native Americans[81]. Samples were retained with Dish QC (DQC) ≥ 0.82. SNP genotypes were first filtered for high-quality cluster differentiation and 95% call rate within batches using SNPolisher. Additional quality control was performed using PLINK. SNPs having an overall call rate less than 95% or discordant calls in duplicate samples were dropped. Samples were dropped for unexpected duplicates in IBD analysis or mismatched sex between genetics and self-report; for first-degree relatives, one sample was retained. All samples had at least 95% genotyping and no evidence of excess heterozygosity (maximum < 2.5*SD). We tested for Hardy-Weinberg Equilibrium (HWE) and cross-batch association for batch effects using a subset of subjects that were of European ancestry and negative for double-stranded-DNA antibodies and renal disease to minimize genetic heterogeneity. SNPs were dropped if HWE $p$ < 1e-5 or any cross-batch association $p$ < 5e-5.

Genetic data was imputed using the Michigan Imputation Server[82] using Minimac3. Imputation was performed using the 1000 Genomes Phase 3 reference panel. Following imputation, SNPs with minor allele frequency greater than 5% were retained, and SNPs with > 5% missing data or evidence of deviation from Hardy Weinberg equilibrium ($p < 1 \times 10^{-4}$) were removed. SNPs were pruned so that no two SNPS were in linkage disequilibrium ($r^2 > 0.8$).

**Phenotypic clustering analysis.** To stratify SLE patients into clinically relevant clusters, we performed unsupervised clustering on patient phenotypic data. Cluster analysis was performed on the American College of Rheumatology (ACR) criteria and sub criteria. Data were dichotomized to represent absence or presence of each criterion. Multiple correspondence analysis was performed with the PCAmixdata R package[83]. The top two MCA dimensions were retained as selected by the k-fold cross validation scheme implemented in the missMDA R package[84,85] (Figure S7A). The number of clusters, k, was chosen by maximizing cluster stability measured by Jaccard similarity using a bootstrap resampling based method. Maximum cluster stability was achieved with $k = 3$ and each cluster had a Jaccard mean stability score greater than 0.82[86] (Fig. S7B).

**Medication use adjustment.** Since medication use can modify CpG methylation at certain sites, we aimed to include medications prescribed at the time of blood sampling as covariates in statistical analyses. We performed principal component analysis (PCA) on a dichotomized matrix of current medications at the time of blood sampling for each patient. The top three PCs were chosen using a three-fold cross validation scheme implemented in the missMDA R package[84,85] and included as covariates in subsequent statistical models.

**Differential methylation analysis.** In order to account for possible confounding due to cell type heterogeneity, we applied the ReFACTor algorithm[87] implemented in Glint[88] to infer peripheral blood cell composition. To identify CpG sites associated with clinical clusters, a linear model adjusted for sex, age, cell count estimates, alcohol use, smoking status, genetic ancestry components, and the top three medication principal components was fit using the nestedF mode in the Limma R package[40]. $P$ values were adjusted using the Benjamini Hochberg procedure. All analyses were performed using R version 3.4.2[89].

**Chromatin state enrichment.** 15-state chromatin model epigenome data for all human peripheral blood cell types was accessed using the NIH Roadmap Epigenomics Consortium[41]. All CpGs on the probe-set were assigned a chromatin state. For each of the 15 chromatin states, a fold statistic was computed using a Fisher's exact test for enrichment of the chromatin state within the set of cluster-associated CpGs relative to all the CpGs in the probe-set. This process was repeated for H3K4me3, H3K4me1, and H3K27ac ChIP seq peaks from the NIH Roadmap Epigenomics Consortium.

**CpG race enrichment adjusted for genetic ancestry.** Enrichment of race-associated CpGs in the list of differentially methylated CpGs was determined via a permutation method. Briefly, the total number of cluster-associated CpGs ($N_{cluster}$) was obtained for a specified FDR as above. Then, a null set was created by randomly permuting the race labels 1000 times. For each permutation, from the set of cluster-associated CpGs, we computed the number of CpGs associated with the permuted race labels by fitting a linear model for each CpG adjusting for sex, age, cell count estimates, alcohol use, smoking status, the top three genetic principal components, and the top three medication principal components. We then found the number of race associated CpGs in the set of cluster-associated CpGs ($N_{race}$) using the same linear model as the null set. We defined an enrichment statistic as the proportion $N_{race}/N_{cluster}$ relative to the mean of the null distribution.

**Statistical meQTL analyses.** Since the Illumina BeadChip EPIC platform is known to cross-react with several probes if the region contains a SNP[90], we first removed all probes with SNPs. meQTL analysis was then performed by fitting a linear model adjusted for sex, age, cell count estimates, alcohol use, smoking status, the top three genetic principal components, and the top three medication principal components using the Matrix eQTL R package[91].

**Causal inference test.** For each meQTL association, the genotype (G)–methylation (M)–cluster (Y) relationships were assessed using the Causal Inference Test (CIT). To establish a mediation relationship in which genotype acts on the clusters through methylation, four conditions must be satisfied: (1) G and Y are associated, (2) G is associated with M after adjusting for Y, (3) M is associated with Y after adjusting for G, and (4) G is independent of Y after adjusting for M. The CIT $p$-value is defined as the maximum of the four component test p-values.

**External validation.** A random forest model was trained on the cluster labels from the study data and cross validation was used to optimize parameters. Methylation data and ACR criteria were obtained from a previously published SLE cohort[12]. All ACR criteria were dichotomized in a manner identical to the study data. The random forest model was then applied to the external validation data to generate cluster labels. CpGs that were differentially methylated in the study data were validated using a linear model adjusted for cell composition, genetic principal components, sex, and age. Smoking history was negative for all patients in the validation cohort. Alcohol use and medication principal components were not included in the linear model since these data were not available.

## Data availability

DNA methylation, genotype and phenotypic data that support the findings of this study have been deposited in DbGap with the primary accession code of phs001850.v1.p1. Data is available through an application to a data access committee

## Code availability

All custom code is available at github.com/ishanparanjpe/lupus_clustering.

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

## Acknowledgements

We would like to gratefully acknowledge the patients who participated in this study. We would also like to thank Noah Zaitlen and Jimmie Ye for reviewing this manuscript and their thoughtful comments. This study was funded through the following grants: P30 AR070155 (M.S., I.P., L.A.C.), P60AR053308 (L.A.C.), K01LM012381 (M.S.), F32 AR070585 NIAMS (M.G.), U01DP005120 CDC (L.A.C., C.M.L., J.Y., M.D., L.T., P.K.), the Rheumatology Research Foundation 128849A (C.M.L.), and the Lupus Research Alliance (L.A.C.).

## Author contributions

C.M.L. and I.P.: data generation, study design, data analysis, data interpretation, and paper preparation. J..N. and K.E.T.: data generation, quality control analyses, genetic ancestry estimates. M.G. and S.A.: study design, data interpretation and revision of paper. M.P.: data interpretation, revision of paper, and creation of the RShiny Application website. S.A.C.: provided data for the validation cohort, data interpretation, and revision of manuscript. B.R.: normalization and quality control analysis for the DNA methylation data. L.F.B.: data generation (DNA methylation profiling). L.T., P.K., M.D., J.Y.: CLUES co-investigators. Patient enrollment and clinical and demographic characterization of patients. M.S.: Computational expertise, study design, data interpretation and paper revisions. L.A.C.: data generation, study design, data interpretation and paper revisions.

## Additional information

**Competing interests:** The authors declare no competing interests.

