## [Peer Review File · Nature Communications]

Reviewers' Comments:

Reviewer #1:

Remarks to the Author:

The authors present a cohort of SLE patients in which they have identified three clusters based on clinical phenotyping data. They identify 256 CpG sites that are differentially methylated across these clusters and replicate 132 of these in a validation cohort. Importantly, this study presents a multi-ethnic derivation cohort which is particularly relevant for SLE given its higher prevalence and greater severity in non-European populations. The shiny app is also an excellent way of making the data accessible to a range of readers. However, I have a number of concerns regarding some of the statistical analyses and presentation of results as outlined below.

Major comments

1. On page 7, cluster S2 is described as the most severe subtype, however not all of the clinical features highlighted in the text as being a marker of this severity appear to support this claim
 - a. In Table 1, if lupus nephritis is represented by "renal" then there is little difference in the prevalence between S1 and S2 (62.3% and 57.7% respectively) and in fact is slightly lower in S2
 - b. The lupus severity index does not look like it has a different distribution between cluster S1 and S2 (Fig 2C).
2. The final paragraph on page 9 states that there are average differences in education level and average income across the clusters but that these differences are not statistically significant. This is a confusing paragraph as the data therefore do not support there being any differences across the clusters.
3. On page 12, the CpG sites hypermethylated in cluster M relative to cluster S1 and S2 are described as suggesting a role for gene silencing in this milder cluster. However, 36/93 of these CpG sites are hypermethylated in the gene body which would suggest 1/3 of these sites are associated with gene expression rather than gene silencing. Furthermore, the next paragraph goes on to identify specific genes (IFI44L, MX1, PARP9, EPSTI1 and PDE7A) which are all hypermethylated in cluster M and are stated as being associated with gene expression rather than silencing. These two paragraphs appear contradictory.
4. The supplementary tables have not been labelled correctly and the legends are not very detailed making it challenging to assess what data are being presented to support the results in the manuscript
 - a. Table S4 appears to have been labelled as Table S8 and vice versa
 - b. Table S5 appears to have been labelled as Table S7 and vice versa
 - c. It is unclear in Table S7 how the SNPs were identified as being associated with clinical cluster (no FDR values are presented). It is also unclear what the "relative risk ratio (relative to cluster M)" is assessing and why MAF for each cluster has been presented when this isn't referenced in the text
5. The volcano plots presented in Figure S2 look very unusual. Many of the methylation sites that have been highlighted as differentially methylated (coloured red, $FDR < 0.1$) have very small effect sizes (delta beta). Also, based on what is described in the text on page 12/13 and Table S5, I would expect 29, 247 and 20 differentially methylated sites to be coloured as significant in the S1vM, S2vM and S1vS2 comparisons respectively whereas many more points on the volcano plot are highlighted as significant.
6. Could the authors comment on the validity and limitations of the validation cohort used, particularly with respect to the ethnicity and sex of the patients
 - a. Given the sex bias in SLE it is understandable that the validation cohort is only female but was

the derivation (CLUES) cohort also only female? As sex was included as a covariate in some of the models it would suggest not but the number of males isn't described anywhere

b. If there were male patients in the CLUES cohort, were they distributed across the original clusters? If not, how might this affect the clusters identified in the validation cohort when only females are analysed?

c. Given that the validation cohort is European only and the severity of SLE is known to vary across ethnic groups, did the validation cohort have a similar range of disease severity as the derivation cohort?

d. How many individuals were assigned to each of the clusters in the validation cohort? Was this similar proportions to the derivation cohort?

7. On page 19, the current permutation analysis may be biased because the null distribution could end up including random CpG sites that do not vary in methylation status across the cohort and therefore would never be associated with self-reported race (or any other variable). This could produce a null distribution with an artificially low number of associations. Instead, it would be helpful to see the results of a permutation analysis where the original 256 CpG sites are maintained and instead self-reported race is shuffled across the samples 1,000 times and the number of associations recorded.

8. Could the authors comment on how much medication use might have an impact on the designation of the original clusters? For example, are S1 and S2 actually one cluster that has been stratified because of medication use. Given that medication use is included in the model for identifying differentially methylated CpGs, could this explain why so few sites are differentially methylated between S1 and S2?

9. The current presentation of the data does not fully support the final conclusion of the paper which states "we have identified 3 distinct clinical subtypes of SLE that can be distinguished by a methylation signature". While three clusters have been identified from the clinical data and CpG sites have been identified which are differentially methylated between them, the analysis does not demonstrate that the clusters can be distinguished from the methylation signature itself.

Minor comments

1. In Table 1 it would be helpful to have the number of individuals in each group as well as %, particularly for ethnic groups

2. How are the criteria significantly associated with cluster calculated in Fig 2A?

3. In the legend for Figure 3, subfigures A and C are described instead of A and B.

4. On page 12, should the citation for Table S5 actually be for Table S4?

5. S4 is missing a figure legend

6. EPTSI1 on page 6 is EPSTI1 in Table S6

7. On page 13, the text states that 20 CpGs are differentially methylated between S1 and S2 but there are only 18 in Table S5

8. On page 17, the citation for Figure 4 should be for Figure 5.

9. Can the 24 meQTL with evidence of mediation of the genetic association by methylation at the corresponding CpG be validated in the validation dataset or was genotyping data not available?

Reviewer #2:

Remarks to the Author:

This is a study describing state of the art genomic and epigenomic analyses of peripheral blood leukocytes from patients with lupus flares of varying severity. The paper is well written and submitted by a leader in this field. The only criticism is that the studies were performed on

unfractionated blood leukocytes, as acknowledged by the author. Lupus flares are characterized by autoimmune cytopenias, and corticosteroids, used to treat lupus flares, also cause lymphopenia and leukocytosis. This confounds the interpretation of the data, so the studies provide limited insights into the biologic processes underlying lupus flares

Reviewer #3:

Remarks to the Author:

This paper by Lanata et al., addresses an important question in SLE, specifically, what is the genetic and molecular basis of the racial disparities in disease course. To do this the authors take a well characterized cohort of SLE patients of different racial backgrounds and perform an unsupervised clustering of these patients using ACR criteria and sub-criteria for SLE. This classifies the cohort into three stable groups, Mild, Severe 1 and Severe 2. Then using this clustering, they evaluate each group for changes in methylation patterns across groups to identify cluster-associated CpG differences and cis-meQTL associations. They then replicate their findings by training an external SLE cohort (with available methylation data) on the original study cohort using a random forest model to obtain similar classification of patients within the replication cohort. Overall, the paper is well written and the analytical approaches are sound. The authorship team is strong and are experts in these types of analyses. While this work has potential there are a number of issues that should be addressed.

Major Issues:

1) The title of the paper is misleading. The title suggests that the integrative omics aspect of the work identified the three classifications of SLE patients. In reality, the clustering was done on ACR criteria and once stable clusters were identified, the omics studies were undertaken to see what was unique about each of the three clusters. This same issue also comes across in the first sentence of the first discussion section paragraph. Did the authors attempt to cluster first on the methylation data and if not, why not?

2) The replication experiment seems flawed. They used a random forest model to apply the cluster classification of the multi-ethnic cohort to a fully European cohort to obtain three classification sets in the replication cohort. The S1 and S2 classification using the study cohort are dominated by non-European subjects which suggests that ethnicity is important in driving these effects. By using an all European validation cohort isn't there some concern about forcing the model derived in the multi-ethnic cohort on to an European cohort? IF the all European cohort falls into the same classification patterns as the multi-ethnic cohort does this suggest that race is not major driving factor in disease severity? This is counter-intuitive of course. What would the validation set look like if the MCA clustering was done on that group up front? Would the three classifications still be there or would this set cluster differently?

3) The study identified only 256 differentially methylated sites, over a third of the methylation sites are in Type 1 IFN response genes. This seems to a rather low number of CpG sites which probably is a result of this study being a case-only study. Given that so many of these are IFN response gene related one has to ask whether this observation is a significant advance in our understanding of the molecular basis of SLE.

4) The overall paper is confusing in its emphasis. The discussion section has several paragraphs describing gene functions but doesn't say much about the race aspects of the paper. There are places where this is touted as a strength and that is definitely true but the message is not coherent and the reader doesn't come away with any better understanding of the role of race-specific differential methylation in human SLE.

Minor points:

1) On line 182-183 a suggestion is made that a role for epigenetic silencing may help explain the

M cluster. This is not accurate statement since you don't really know what "normal" is in this experiment. You can only say this relative to the S1 and S2 clusters so that should be added to the text.

2) On line 86 the sentence makes it seem that previous studies have identified methylation differences in HLA-DR2 and HLA-DR3. These are actually names of HLA haplotypes not specific genes. This should be revised.

3) The word causal appears 6 times and the word casual appears 5 times. This needs to be fixed.

4) Supplemental tables 1-3 are referred to in the paper but not uploaded.

Authors' response.

We thank the reviewers for the thoughtful comments and important points regarding our manuscript. We have considered all of these points and addressed them below and in our revised manuscript. Our point-by-point response are outlined below. Overall, we have made the following major changes to the manuscript to address the central areas of concern of the reviewers:

- 1) The main title of the paper was changed to better reflect our analytic approach.
- 2) The discussion was reframed to better reflect the main take-away points of our findings. By studying a multi-ethnic SLE cohort, we were able to identify 3 distinct clinical clusters, establishing a framework to study disease heterogeneity in SLE. These clinical clusters had differentially methylated CpGs in relevant immune pathways. A minority of these were under the influence of genetics, and we described interesting genetic-methylation associations. However, the majority of these methylation marks were associated with self-reported ethnicity after adjusting for genetics, suggesting that methylation may be a molecular biomarker of unmeasured environmental effects that are present in distinct ethnic groups.
- 3) We clarified important points regarding our validation cohort which resulted in contradictory messages in our original manuscript. Although we could validate the clusters in a cohort of patients of European descent, the proportions of patients that fell under these 3 clusters differed, reflecting the different ethnic distribution between the cohorts. Regarding the methylation validation, we were limited by the different methylation arrays utilized, therefore we could only test 61% of the CpGs that were significant in the primary multiethnic cohort. Furthermore, we did not have genetic data to perform similar analyses in the validation cohort, therefore we were unable to further analyze the genetic and non-genetic effects in methylation. Despite these limitations, we could stratify patients in 3 clinical clusters, and validate 80% of the tested CpGs, demonstrating feasibility and reproducibility of our findings.

We believe that this paper represents a novel approach for studying disease heterogeneity in SLE. In addition, it highlights the importance of studying patients of multiple ethnicities simultaneously in order to better study the genomic drivers of disease outcomes and ethnic disparities.

Reviewers' comments:

Reviewer #1 (Remarks to the Author):

The authors present a cohort of SLE patients in which they have identified three clusters based on clinical phenotyping data. They identify 256 CpG sites that are differentially methylated across these clusters and replicate 132 of these in a validation cohort. Importantly, this study presents a multi-ethnic derivation cohort which is particularly relevant for SLE given its higher prevalence and greater severity in non-European populations. The shiny app is also an excellent way of making the data accessible to a range of readers. However, I have a number of concerns regarding some of the statistical analyses and presentation of results as outlined below.

Thank you for a nice summary of the study and for recognizing the value of the presented work.

Major comments

1. On page 7, cluster S2 is described as the most severe subtype, however not all of the clinical features highlighted in the text as being a marker of this severity appear to support this claim

a. In Table 1, if lupus nephritis is represented by “renal” then there is little difference in the prevalence between S1 and S2 (62.3% and 57.7% respectively) and in fact is slightly lower in S2

b. The lupus severity index does not look like it has a different distribution between cluster S1 and S2 (Fig 2C).

We apologize for any confusion here, but the interpretation of clusters S1 and S2 deserves clarification. S1 and S2 both represent severe subtypes, as evidenced by greater lupus severity index as compared to cluster M. Although there is not a significant difference in lupus nephritis incidence between clusters S1 and S2, cluster M has a significantly lower incidence. S1 and S2 can be distinguished by clinically significant differences in rates of leukopenia, lymphopenia, photosensitivity, pleuritis, malar rash and discoid rash and more autoantibody production. We considered S2 a more severe cluster as there was a higher proportion of ACR criteria as well as a higher proportion of different autoantibodies which tend to correlate with disease manifestations. Since the lupus severity index is a gross metric of disease severity that incorporates multiple ACR criteria, this index does not represent the clinical differences observed between clusters S1 and S2. We have clarified this point in the text on page 7 (tracked changes document).

2. The final paragraph on page 9 states that there are average differences in education level and average income across the clusters but that these differences are not statistically significant. This is a confusing paragraph as the data therefore do not support there being any differences across the clusters.

We apologize for the confusion and have changed this paragraph as below:

In a comparison of socioeconomic variables across clinical clusters, we did not observe a statically significant difference in average education level or income between the three clusters (Table S3).

3. On page 12, the CpG sites hypermethylated in cluster M relative to cluster S1 and S2 are described as suggesting a role for gene silencing in this milder cluster. However, 36/93 of these CpG sites are hypermethylated in the gene body which would suggest 1/3 of these sites are associated with gene expression rather gene silencing. Furthermore, the next paragraph goes on to identify specific genes (IFI44L, MX1, PARP9, EPSTI1 and PDE7A) which are all hypermethylated in cluster M and are stated as being associated with gene expression rather than silencing. These two paragraphs appear contradictory.

We thank the reviewer for pointing out this important discrepancy. We have modified the results section to discuss differences in methylation in the gene body and upstream sites. CpGs hypermethylated in the promoter region (TSS200, TSS1500, 5'UTR) have been described as silencing and CpGs hypermethylated in the gene body have been described as activating:

Notably, of the 101 IFN-alpha CpGs, 93 were hypermethylated in cluster M relative to both cluster S1 and S2,. Of these CpGs, 57 were in the promoter region (TSS200, TSS1500, 5' UTR), and 36 were in the gene body. Hypermethylation at the promoter sites suggests a role for epigenetic silencing in cluster M with respect to S1 and S2 while gene body hypermethylation suggest gene expression.

Cluster-associated CpGs with the greatest variance (5-11% methylation variance) across the clusters were in genes IFI44L, MX1, PARP9, EPSTI1 and PDE7A, all displaying hypermethylation in cluster M relative to S1 and S2 (Table S5). With the exception of PDE7A, all of these genes are interferon responsive. PDE7A encodes a phosphodiesterase associated with T cell activation and IL-2 production (39). Differentially methylated CpGs in IFI44L, MX1 and PARP9 map to the 5-UTR region,

suggesting silencing of these genes. Differentially methylated CpGs in EPSTI1 and PDE7A are located in the gene body, where hypermethylation is associated with gene expression.

4. The supplementary tables have not been labelled correctly and the legends are not very detailed making it challenging to assess what data are being presented to support the results in the manuscript

a. Table S4 appears to have been labelled as Table S8 and vice versa

b. Table S5 appears to have been labelled as Table S7 and vice versa

c. It is unclear in Table S7 how the SNPs were identified as being associated with clinical cluster (no FDR values are presented). It is also unclear what the “relative risk ratio (relative to cluster M)” is assessing and why MAF for each cluster has been presented when this isn’t referenced in the text

We apologize for any confusion here and have corrected the labeling for a and b. Since Table S7 is not relevant to our core analysis, we have removed it for clarity (c).

5. The volcano plots presented in Figure S2 look very unusual. Many of the methylation sites that have been highlighted as differentially methylated (coloured red, $FDR < 0.1$) have very small effect sizes ($\Delta\beta$). Also, based on what is described in the text on page 12/13 and Table S5, I would expect 29, 247 and 20 differentially methylated sites to be coloured as significant in the S1vM, S2vM and S1vS2 comparisons respectively whereas many more points on the volcano plot are highlighted as significant.

We apologize for this error and have recreated the volcano plots to be consistent with the results presented in Table S5. Red points indicate CpGs with adj. p value < 0.05 and blue points indicate CpGs with adj. p value < 0.05 and $DBeta > 0.05$.

6. Could the authors comment on the validity and limitations of the validation cohort used, particularly with respect to the ethnicity and sex of the patients.

We have added a paragraph to the discussion section (page 27 and 28) to further discuss our validation cohort. In particular we have added the following:

“We applied this approach to an external cohort of exclusively SLE patients of European descent. The clinical clusters were labelled by applying a random forest model trained on the original CLUES clusters. The greatest number of subjects were assigned to cluster M. Since cluster M was also enriched for patients that self-identified as White in the original clusters, we believe that the clustering model accurately captures the role of racial differences in lupus severity. We were only able to test 158 of the 256 cluster-associated CpGs in the validation cohort due to a limited number of overlapping probes. However, we validated 84% of the cluster-associated methylation sites. As we also found that the majority of cluster-associated CpGs were race-associated in the CLUES cohort, this suggests that while race plays a role in DNA methylation, these effects are secondary to the clinical differences between lupus clusters. As we could not determine the relative genetic and non-genetic contributions of the methylation differences, we cannot recreate the race-specific findings we have described for the CLUES cohort. In general, this suggests that the subtypes and associated methylation differences we have described can be applied to other cohorts”

“Additionally, our validation cohort is limited not only by its ethnic and gender composition, but also in methylation as it was measured on a different chip than the present study”.

a. Given the sex bias in SLE it is understandable that the validation cohort is only female but was the derivation (CLUES) cohort also only female? As sex was included as a covariate in some of the models it would suggest not but the number of males isn’t described anywhere

- b. If there were male patients in the CLUES cohort, were they distributed across the original clusters? If not, how might this affect the clusters identified in the validation cohort when only females are analysed?
- c. Given that the validation cohort is European only and the severity of SLE is known to vary across ethnic groups, did the validation cohort have a similar range of disease severity as the derivation cohort?
- d. How many individuals were assigned to each of the clusters in the validation cohort? Was this similar proportions to the derivation cohort?

- a. We have updated Table S1 to include the distribution of sex in the CLUES cohort. The CLUES cohort included 298 females and 37 males.
- b. The sex distribution was not significantly different across the original clusters ($p = 0.94$). Cluster M was 89.1% female, cluster S1 was 88.3% females and cluster S2 was 89.7% female. The number of female subjects in each cluster has been added to Table S1.
- c. The CLUES data had a significantly greater lupus severity index (6.85 ± 1.63 in CLUES and 6.15 ± 1.42 in validation; $p < 0.001$), however this difference is relatively small and not clinically significant. Although a cohort with matched disease severity index would be an ideal validation dataset, we were limited by availability of data.
- d. In the validation data set, 164 (47%) were assigned cluster M, 114 (35%) were assigned cluster S1 and 58 (18%) were assigned cluster S2. In comparison to the CLUES cohort where the majority of subjects were in cluster S1, in the validation dataset, the majority were in cluster M. This may be due to the fact that in the cluster derivation (Table 1), cluster M was enriched for European subjects and the validation dataset is entirely European. This was included in the results section (page 14) as well as in the discussion section (page 28).

7. On page 19, the current permutation analysis may be biased because the null distribution could end up including random CpG sites that do not vary in methylation status across the cohort and therefore would never be associated with self-reported race (or any other variable). This could produce a null distribution with an artificially low number of associations. Instead, it would be helpful to see the results of a permutation analysis where the original 256 CpG sites are maintained and instead self-reported race is shuffled across the samples 1,000 times and the number of associations recorded.

Thank you for the suggestion. We have repeated the permutation analysis by shuffling self-reported race across the samples 1000 times as described. Using this method, we still observe a significantly greater number of race-associated CpGs in our 256 CpG signature over the null distribution ($p < 0.001$). Figure 6A has been updated to display the results of this new analysis.

8. Could the authors comment on how much medication use might have an impact on the designation of the original clusters? For example, are S1 and S2 actually one cluster that has been stratified because of medication use. Given that medication use is included in the model for identifying differentially methylated CpGs, could this explain why so few sites are differentially methylated between S1 and S2?

Thank you for raising this important point. It is possible that the ACR phenotypes used for clustering were affected by medication use. Ideally, our study would include patients with a similar treatment history to minimize the uncontrolled effect of medication on organ damage and lymphopenia. At the time of blood draw, we only found a significant difference in mycophenolate and prednisone use between the clusters, with lower use in cluster M as compared to S1 and S2. Thus, it is possible that by controlling for medication use, we fail to identify CpGs whose methylation is correlated with medication use. However, adjusting by medication use is necessary to ensure that the methylation

associations we report are not false positives driven only by medications. This represents a limitation of our study and future work may use more stringent inclusion criteria to limit effect of medication confounding.

9. The current presentation of the data does not fully support the final conclusion of the paper which states “we have identified 3 distinct clinical subtypes of SLE that can be distinguished by a methylation signature”. While three clusters have been identified from the clinical data and CpG sites have been identified which are differentially methylated between them, the analysis does not demonstrate that the clusters can be distinguished from the methylation signature itself.

We apologize for the inaccurate wording here. We have removed this interpretation from the first paragraph of the discussion section and modified it as follows:

We also show that patterns of differential methylation at specific CpGs are associated with the clinical subtypes. A subset of these CpGs are under genetic control and display a strong association with self-reported race after adjusting for genetic ancestry.

Minor comments

1. In Table 1 it would be helpful to have the number of individuals in each group as well as %, particularly for ethnic groups

In Table 1, the number of individuals in each group is given at the top. The numbers for each racial group are percentages within each cluster.

2. How are the criteria significantly associated with cluster calculated in Fig 2A?

Association between each criterion and cluster was evaluated by a Fisher exact test. This has been added to the figure caption.

3. In the legend for Figure 3, subfigures A and C are described instead of A and B.

Sorry about this, the error has been corrected.

4. On page 12, should the citation for Table S5 actually be for Table S4?

It is Table S5 and has been corrected. We apologize for the error.

5. S4 is missing a figure legend

A figure legend has been added.

6. EPTSI 1 on page 6 is EPSTI 1 in Table S6

Thank you for pointing this out. We changed EPTSI1 to EPSTI1 in the text.

7. On page 13, the text states that 20 CpGs are differentially methylated between S1 and S2 but there are only 18 in Table S5

Thank you for pointing this out. We have modified the text to state that 18 CpGs were differentially methylated between S1 and S2.

8. On page 17, the citation for Figure 4 should be for Figure 5.

Sorry about the error - we have changed the citation to Figure 5.

9. Can the 24 meQTL with evidence of mediation of the genetic association by methylation at the corresponding CpG be validated in the validation dataset or was genotyping data not available?

Unfortunately genotyping data was not available so we were not able to validate the methylation mediation analysis.

Reviewer #2 (Remarks to the Author):

This is a study describing state of the art genomic and epigenomic analyses of peripheral blood leukocytes from patients with lupus flares of varying severity. The paper is well written and submitted by a leader in this field. The only criticism is that the studies were performed on unfractionated blood leukocytes, as acknowledged by the author. Lupus flares are characterized by autoimmune cytopenias, and corticosteroids, used to treat lupus flares, also cause lymphopenia and leukocytosis. This confounds the interpretation of the data, so the studies provide limited insights into the biologic processes underlying lupus flares.

We would like to thank the reviewer for recognizing the value of the presented study. We agree that we were unable to study the biology of lupus flares with the current study design. As shown in table 1, the disease activity score SLEDAI was overall low in the three clusters, however, our main point was to characterize disease heterogeneity. We agree that it would be interesting to look at specific cell subsets and we mention this in the limitation section in the discussion. We found that the subjects in cluster S1 and S2 had a significantly lower leukocyte count at time of blood draw as compared to cluster M (Table S2). Since we adjust by cell proportions in the differential methylation analysis, our analysis may fail to identify methylation differences that are correlated with cell proportion. We have added a few sentences to the discussion section describing this as future work.

We acknowledge that we were not able to capture a significant number of patients with active disease manifestations. Future studies may include stringent inclusion criteria to rigorously control for covariates that contribute to lupus flares. We believe the strength of our study is unbiased discovery of clinical subtypes and association with methylation differences. Since we adjust by medication use in our differential methylation analysis, these methylation differences are likely not driven by medication use alone and represent true biological differences underlying clinical subtypes.

Reviewer #3 (Remarks to the Author):

This paper by Lanata et al., addresses an important question in SLE, specifically, what is the genetic and molecular basis of the racial disparities in disease course. To do this the authors take a well characterized cohort of SLE patients of different racial backgrounds and perform an unsupervised clustering of these patients using ACR criteria and sub-criteria for SLE. This classifies the cohort into three stable groups, Mild, Severe 1 and Severe 2. Then using this clustering, they evaluate each group for changes in methylation patterns across groups to identify cluster-associated CpG differences and cis-meQTL associations. They

then replicate their findings by training an external SLE cohort (with available methylation data) on the original study cohort using a random forest model to obtain similar classification of patients within the replication cohort. Overall, the paper is well written and the analytical approaches are sound. The authorship team is strong and are experts in these types of analyses. While this work has potential there are a number of issues that should be addressed.

Thank you for recognizing the value of the presented study. We include the point by point responses below.

Major Issues:

1) The title of the paper is misleading. The title suggests that the integrative omics aspect of the work identified the three classifications of SLE patients. In reality, the clustering was done on ACR criteria and once stable clusters were identified, the omics studies were undertaken to see what was unique about each of the three clusters. This same issue also comes across in the first sentence of the first discussion section paragraph. Did the authors attempt to cluster first on the methylation data and if not, why not?

We apologize for this confusion. We have revised the title of the paper to be: "Subtyping Systemic Lupus Erythematosus: a phenotypic and genomics approach in a multi-ethnic cohort"

We attempted to cluster the DNA methylation data independently, however we did not find stable clusters. Moreover, due to the strong association of DNA methylation with genetic variation, unsupervised clustering of the methylation data revealed population structure rather than lupus-relevant clinical differences. For these reasons, we did not report DNA methylation clustering and rather chose to define subtypes by ACR criteria. We have added this in the discussion section (page 22-23)

2) The replication experiment seems flawed. They used a random forest model to apply the cluster classification of the multi-ethnic cohort to a fully European cohort to obtain three classification sets in the replication cohort. The S1 and S2 classification using the study cohort are dominated by non-European subjects which suggests that ethnicity is important in driving these effects. By using an all European validation cohort isn't there some concern about forcing the model derived in the multi-ethnic cohort on to an European cohort? IF the all European cohort falls into the same classification patterns as the multi-ethnic cohort does this suggest that race is not major driving factor in disease severity? This is counter-intuitive of course. What would the validation set look like if the MCA clustering was done on that group up front? Would the three classifications still be there or would this set cluster differently?

Thank you for raising this important point. We agree that the validation cohort is not ideal in terms of the demographics distribution and point this out in the limitation section in the discussion. However, in this study, our aim was to validate the differential methylation analysis, rather than a comprehensive validation of the clinical subtypes which would require a larger, well-controlled dataset.

In the CLUES study cohort, the majority of subjects were in the "S1" cluster. As the reviewer pointed out, race plays a role in driving lupus severity, so the European validation cohort should have a different distribution of clusters. Indeed, when the random forest model was applied to the validation data, cluster M had the greatest number of subjects (164), followed by S2 (86) and S1 (76). Since patients of European descent tend to have less severe disease, our random forest model robustly classified the majority of the validation cohort patients into the less severe cluster. Thus, even though

an ideal cohort would have a similar distribution of race, we believe our random forest method accurately classifies the validation cohort subjects and allows for us to perform the downstream differential methylation analysis.

When MCA clustering was performed separately on the validation cohort, we also observed three stable clusters (Jaccard score >0.8). Similar to the CLUES study clustering, one cluster was mild with a lower lupus severity index and two clusters were more severe with a greater lupus severity index (Figure 1). We have included an additional supplementary table (Table S6) that presents the clinical characteristics and number of subjects in each cluster for the validation cohort.

Figure 1 Comparison of validation cohort clustering methods. On the left, the lupus severity index of clusters generated by applying the CLUES study random forest model is presented. On the right, the lupus severity index of clusters generated by applying MCA clustering on the validation data is presented. MCA clustering was applied using the first two principal components and cluster stability was evaluated using the Jaccard score consistent with the original study clustering method.

3) The study identified only 256 differentially methylated sites, over a third of the methylation sites are in Type 1 IFN response genes. This seems to a rather low number of CpG sites which probably is a result of this study being a case-only study. Given that so many of these are IFN response gene related one has to ask whether this observation is a significant advance in our understanding of the molecular basis of SLE.

Thank you for raising this important point. We have performed a rigorous methylation analysis, adjusting for multiple confounders that have likely contributed to a small number of methylation sites. We were interested in studying disease heterogeneity and outcomes, hence the focus on a case-only study design. Our study is novel in several respects. This is the first EWAS that includes 4 ethnic groups, more closely reflecting the true SLE population. Although it is known that SLE and SLE severity are associated with upregulation of interferon responsive genes, in this approach we are able to determine that the upregulation of IFN-related genes is not only associated with disease severity but also associated with self-reported ethnicity and much less to genetic variation.

It is possible that methylation is a result of lupus-related organ damage or cellular response rather than a driver of disease progression. The hypothesis presented, namely that patterns of methylation distinguishes SLE clinical subtypes, must be validated in mechanistic studies in future studies. Nonetheless, it represents a significant advance in terms of the biological basis of SLE.

4) The overall paper is confusing in its emphasis. The discussion section has several paragraphs describing gene functions but doesn't say much about the race aspects of the

paper. There are places where this is touted as a strength and that is definitely true but the message is not coherent and the reader doesn't come away with any better understanding of the role of race-specific differential methylation in human SLE.

Thank you for the comment. We have restructured the discussion so that the messages from the paper are clear. There are three main points. We were able to find 3 distinct clinical clusters which had differences in DNA methylation. Of the methylation changes that were found, a subset had evidence of genetic influence and the majority did not. We discussed interesting genetic-methylation associations, as hypotheses for disease pathogenesis. We then discussed the methylation changes that were not under genetic control, however strongly associated with self-reported race. In this section of the discussion we argue that we are likely capturing the molecular mediators of environmental effects associated with different ethnicities, highlighting methylation as a potential molecular mediator of the race effect in SLE disparities.

Minor points:

1) On line 182-183 a suggestion is made that a role for epigenetic silencing may help explain the M cluster. This is not accurate statement since you don't really know what "normal" is in this experiment. You can only say this relative to the S1 and S2 clusters so that should be added to the text.

Thank you for raising this point - we have modified the section as below:

Notably, of the 101 IFN-alpha CpGs, 93 were hypermethylated in cluster M relative to both cluster S1 and S2. Of these CpGs, 57 were in the promoter region (TSS200, TSS1500, 5' UTR), and 36 were in the gene body. Hypermethylation at the promoter sites suggests a role for epigenetic silencing in cluster M with respect to S1 and S2 while gene body hypermethylation suggest gene expression.

2) On line 86 the sentence makes it seem that previous studies have identified methylation differences in HLA-DR2 and HLA-DR3. These are actually names of HLA haplotypes not specific genes. This should be revised.

We apologize for the error - the wording has been modified as below:

There is evidence that both genetics and DNA methylation play a role in SLE outcomes. Lupus nephritis, a severe outcome of lupus that drives disease mortality, was found to be significantly correlated with genetic variants in ITGAM, TNFSF4, APOL1, PDGFRA, SLC5A11 among others. The HLA-DR2 and HLA-DR3 alleles have also been associated with susceptibility and autoantibody production in lupus.

3) The word causal appears 6 times and the word casual appears 5 times. This needs to be fixed.

We apologize for the oversight - this has been corrected.

4) Supplemental tables 1-3 are referred to in the paper but not uploaded.

These tables were included in the supplementary material PDF and not as separate excel files.

Reviewers' Comments:

Reviewer #1:

Remarks to the Author:

The authors have addressed all of my comments and I am satisfied with the updates they have made to the manuscript.

Reviewer #3:

Remarks to the Author:

This manuscript is significantly improved. I have no further concerns.